# Fighting Bandits with a New Kind of Smoothness

**Jacob Abernethy**
University of Michigan
jabernet@umich.edu

**Chansoo Lee**
University of Michigan
chansool@umich.edu

**Ambuj Tewari**
University of Michigan
tewaria@umich.edu

## Abstract

We provide a new analysis framework for the adversarial multi-armed bandit problem. Using the notion of convex smoothing, we define a novel family of algorithms with minimax optimal regret guarantees. First, we show that regularization via the *Tsallis entropy*, which includes EXP3 as a special case, matches the $O(\sqrt{NT})$ minimax regret with a smaller constant factor. Second, we show that a wide class of perturbation methods achieve a near-optimal regret as low as $O(\sqrt{NT \log N})$, as long as the perturbation distribution has a bounded hazard function. For example, the Gumbel, Weibull, Frechet, Pareto, and Gamma distributions all satisfy this key property and lead to near-optimal algorithms.

## 1 Introduction

The classic *multi-armed bandit* (MAB) problem, generally attributed to the early work of Robbins (1952), poses a generic online decision scenario in which an agent must make a sequence of choices from a fixed set of options. After each decision is made, the agent receives some feedback in the form of a loss (or gain) associated with her choice, but no information is provided on the outcomes of alternative options. The agent's goal is to minimize the total loss over time, and the agent is thus faced with the balancing act of both experimenting with the menu of choices while also utilizing the data gathered in the process to improve her decisions. The MAB framework is not only mathematically elegant, but useful for a wide range of applications including medical experiments design (Gittins, 1996), automated poker playing strategies (Van den Broeck et al., 2009), and hyperparameter tuning (Pacula et al., 2012).

Early MAB results relied on stochastic assumptions (e.g., IID) on the loss sequence (Auer et al., 2002; Gittins et al., 2011; Lai and Robbins, 1985). As researchers began to establish non-stochastic, *worst-case* guarantees for sequential decision problems such as *prediction with expert advice* (Littlestone and Warmuth, 1994), a natural question arose as to whether similar guarantees were possible for the bandit setting. The pioneering work of Auer, Cesa-Bianchi, Freund, and Schapire (2003) answered this in the affirmative by showing that their algorithm EXP3 possesses nearly-optimal regret bounds with matching lower bounds. Attention later turned to the bandit version of *online linear optimization*, and several associated guarantees were published the following decade (Abernethy et al., 2012; Dani and Hayes, 2006; Dani et al., 2008; Flaxman et al., 2005; McMahan and Blum, 2004).

Nearly all proposed methods have relied on a particular algorithmic blueprint; they reduce the bandit problem to the full-information setting, while using randomization to make decisions and to *estimate* the losses. A well-studied family of algorithms for the full-information setting is *Follow the Regularized Leader* (FTRL), which optimizes the objective function of the following form:

$$\arg\min_{x \in \mathcal{K}} L^\top x + \lambda R(x) \tag{1}$$

where $\mathcal{K}$ is the decision set, $L$ is (an estimate of) the cumulative loss vector, and $R$ is a *regularizer*, a convex function with suitable curvature to stabilize the objective. The choice of regularizer $R$ is

critical to the algorithm's performance. For example, the EXP3 algorithm (Auer, 2003) regularizes with the *entropy function* and achieves a nearly optimal regret bound when $\mathcal{K}$ is the probability simplex. For a general convex set, however, other regularizers such as *self-concordant barrier functions* (Abernethy et al., 2012) have tighter regret bounds.

Another class of algorithms for the full information setting is *Follow the Perturbed Leader* (FTPL) (Kalai and Vempala, 2005) whose foundations date back to the earliest work in adversarial online learning (Hannan, 1957). Here we choose a distribution $\mathcal{D}$ on $\mathbb{R}^N$, sample a random vector $Z \sim \mathcal{D}$, and solve the following *linear* optimization problem

$$\arg\min_{x \in \mathcal{K}} (L + Z)^\top x. \tag{2}$$

FTPL is computationally simpler than FTRL due to the linearity of the objective, but it is analytically much more complex due to the randomness. For every different choice of $\mathcal{D}$, an entirely new set of techniques had to be developed (Devroye et al., 2013; Van Erven et al., 2014). Rakhlin et al. (2012) and Abernethy et al. (2014) made some progress towards unifying the analysis framework. Their techniques, however, are limited to the full-information setting.

In this paper, we propose a new analysis framework for the multi-armed bandit problem that unifies the regularization and perturbation algorithms. The key element is a new kind of smoothness property, which we call *differential consistency*. It allows us to generate a wide class of both optimal and near-optimal algorithms for the adversarial multi-armed bandit problem. We summarize our main results:

1. We show that regularization via the *Tsallis entropy* leads to the state-of-the-art adversarial MAB algorithm, matching the minimax regret rate of Audibert and Bubeck (2009) with a tighter constant. Interestingly, our algorithm fully generalizes EXP3.

2. We show that a wide array of well-studied noise distributions lead to near-optimal regret bounds (matching those of EXP3). Furthermore, our analysis reveals a strikingly simple and appealing sufficient condition for achieving $O(\sqrt{T})$ regret: the *hazard rate* function of the noise distribution must be bounded by a constant. We conjecture that this requirement is in fact both necessary and sufficient.

## 2 Gradient-Based Prediction Algorithms for the Multi-Armed Bandit

Let us now introduce the adversarial multi-armed bandit problem. On each round $t = 1, \ldots, T$, a learner must choose a distribution $p_t \in \Delta_N$ over the set of $N$ available actions. The adversary (Nature) chooses a vector $g_t \in [-1, 0]^N$ of losses, the learner samples $i_t \sim p_t$, and plays action $i_t$. After selecting this action, the learner observes only the value $g_{t,i_t}$, and receives no information as to the values $g_{t,j}$ for $j \neq i_t$. This limited information feedback is what makes the bandit problem much more challenging than the full-information setting in which the entire $g_t$ is observed.

The learner's goal is to minimize the *regret*. Regret is defined to be the difference in the realized loss and the loss of the best fixed action in hindsight:

$$\text{Regret}_T := \max_{i \in [N]} \sum_{t=1}^{T} (g_{t,i} - g_{t,i_t}). \tag{3}$$

To be precise, we consider the *expected* regret, where the expectation is taken with respect to the learner's randomization.

**Loss vs. Gain Note:** We use the term "loss" to refer to $g$, although the maximization in (3) would imply that $g$ should be thought of as a "gain" instead. We use the former term, however, as we impose the assumption that $g_t \in [-1, 0]^N$ throughout the paper.

### 2.1 The Gradient-Based Algorithmic Template

Our results focus on a particular algorithmic template described in Framework 1, which is a slight variation of the Gradient Based Prediction Algorithm (GBPA) of Abernethy et al. (2014). Note that

the algorithm (i) maintains an *unbiased estimate* of the cumulative losses $\hat{G}_t$, (ii) updates $\hat{G}_t$ by adding a single round estimate $\hat{g}_t$ that has only one non-zero coordinate, and (iii) uses the gradient of a convex function $\tilde{\Phi}$ as sampling distribution $p_t$. The choice of $\tilde{\Phi}$ is flexible but $\tilde{\Phi}$ must be a differentiable convex function and its derivatives must always be a probability distribution.

Framework 1 may appear restrictive but it has served as the basis for much of the published work on adversarial MAB algorithms (Auer et al., 2003; Kujala and Elomaa, 2005; Neu and Bartók, 2013). First, the GBPA framework essentially encompasses all FTRL and FTPL algorithms (Abernethy et al., 2014), which are the core techniques not only for the full information settings, but also for the bandit settings. Second, the estimation scheme ensures that $\hat{G}_t$ remains an unbiased estimate of $G_t$. Although there is some flexibility, any unbiased estimation scheme would require some kind of inverse-probability scaling—information theory tells us that the unbiased estimates of a quantity that is observed with only probabilty $p$ must necessarily involve fluctuations that scale as $O(1/p)$.

---

**Framework 1:** Gradient-Based Prediction Alg. (GBPA) Template for Multi-Armed Bandit

---

GBPA($\tilde{\Phi}$): $\tilde{\Phi}$ is a differentiable convex function such that $\nabla\tilde{\Phi} \in \Delta^N$ and $\nabla_i\tilde{\Phi} > 0$ for all $i$.
Initialize $\hat{G}_0 = 0$
**for** *t = 1 to T* **do**
> **Nature:** A loss vector $g_t \in [-1, 0]^N$ is chosen by the Adversary
> **Sampling:** Learner chooses $i_t$ according to the distribution $p(\hat{G}_{t-1}) = \nabla\Phi_t(\hat{G}_{t-1})$
> **Cost:** Learner "gains" loss $g_{t,i_t}$
> **Estimation:** Learner "guesses" $\hat{g}_t := \frac{g_{t,i_t}}{p_{i_t}(\hat{G}_{t-1})}\mathbf{e}_{i_t}$
>
> **Update:** $\hat{G}_t = \hat{G}_{t-1} + \hat{g}_t$

---

**Lemma 2.1.** *Define* $\Phi(G) \equiv \max_i G_i$ *so that we can write the expected regret of GBPA($\tilde{\Phi}$) as*

$$\mathbb{E}\text{Regret}_T = \Phi(G_T) - \sum_{t=1}^T \langle \nabla\tilde{\Phi}(\hat{G}_{t-1}), g_t \rangle.$$

*Then, the expected regret of the GBPA($\tilde{\Phi}$) can be written as:*

$$\mathbb{E}\text{Regret}_T \leq \underbrace{\tilde{\Phi}(0) - \Phi(0)}_{\textit{overestimation penalty}} + \mathbb{E}_{i_1,\ldots,i_{t-1}}\left[\underbrace{\Phi(\hat{G}_T) - \tilde{\Phi}(\hat{G}_T)}_{\textit{underestimation penalty}} + \sum_{t=1}^T \underbrace{\mathbb{E}_{i_t}[D_{\tilde{\Phi}}(\hat{G}_t, \hat{G}_{t-1})|\hat{G}_{t-1}]}_{\textit{divergence penalty}}\right],$$

(4)

*where the expectations are over the sampling of* $i_t$.

*Proof.* Let $\tilde{\Phi}$ be a valid convex function for the GBPA. Consider GBPA($\tilde{\Phi}$) being run on the loss sequence $g_1, \ldots, g_T$. The algorithm produces a sequence of estimated losses $\hat{g}_1, \ldots, \hat{g}_T$. Now consider GBPA-NE($\tilde{\Phi}$), which is GBPA($\tilde{\Phi}$) run with the full information on the deterministic loss sequence $\hat{g}_1, \ldots, \hat{g}_T$ (there is no estimation step, and the learner updates $\hat{G}_t$ directly). The regret of this run can be written as

$$\Phi(\hat{G}_T) - \sum_{t=1}^T \langle \nabla\tilde{\Phi}(\hat{G}_{t-1}), \hat{g}_t \rangle,$$

and $\Phi(G_T) \leq \Phi(\hat{G}_T)$ by the convexity of $\Phi$. Hence, it suffices to show that the GBPA-NE($\tilde{\Phi}$) has regret at most the righthand side of Equation 4, which is a fairly well-known result in online learning literature; see, for example, (Cesa-Bianchi and Lugosi, 2006, Theorem 11.6) or (Abernethy et al., 2014, Section 2). For completeness, we included the full proof in Appendix A. $\square$

## 2.2 A New Kind of Smoothness

What has emerged as a guiding principle throughout machine learning is that enforcing *stability* of an algorithm can often lead immediately to performance guarantees—that is, small modifications of the input data should not dramatically alter the output. In the context of GBPA, algorithmic stability is guaranteed as long as the dervative $\nabla\tilde{\Phi}(\cdot)$ is Lipschitz. Abernethy et al. (2014) explored a set of conditions on $\nabla^2\tilde{\Phi}(\cdot)$ that lead to optimal regret guarantees for the full-information setting. Indeed,

this work discussed different settings where the regret depends on an upper bound on either the nuclear norm or the operator norm of this hessian.

In short, regret in the full information setting relies on the *smoothness* of the choice of $\tilde{\Phi}$. In the bandit setting, however, merely a uniform bound on the magnitude of $\nabla^2 \tilde{\Phi}$ is insufficient to guarantee low regret; the regret (Lemma 2.1) involves terms of the form $D_{\tilde{\Phi}}(\hat{G}_{t-1} + \hat{g}_t, \hat{G}_{t-1})$, where the incremental quantity $\hat{g}_t$ can scale as large as *the inverse of the smallest probability* of $p(\hat{G}_{t-1})$. What is needed is a stronger notion of the smoothness that bounds $\nabla^2 \tilde{\Phi}$ in correspondence with $\nabla \tilde{\Phi}$, and we propose the following definition:

**Definition 2.2** (Differential Consistency)**.** *For constants $\gamma, C > 0$, we say that a convex function $\tilde{\Phi}(\cdot)$ is $(\gamma, C)$-differentially-consistent if for all $G \in (-\infty, 0]^N$,*

$$\nabla_{ii}^2 \tilde{\Phi}(G) \leq C(\nabla_i \tilde{\Phi}(G))^\gamma.$$

We now prove a useful bound that emerges from differential consistency, and in the following two sections we shall show how this leads to regret guarantees.

**Theorem 2.3.** *Suppose $\tilde{\Phi}$ is $(\gamma, C)$-differentially-consistent for constants $C, \gamma > 0$. Then divergence penalty at time $t$ in Lemma 2.1 can be upper bounded as:*

$$\mathbb{E}_{i_t}[D_{\tilde{\Phi}}(\hat{G}_t, \hat{G}_{t-1})|\hat{G}_{t-1}] \leq C \sum_{i=1}^{N} \left( \nabla_i \tilde{\Phi}(\hat{G}_{t-1}) \right)^{\gamma-1}.$$

*Proof.* For the sake of clarity, we drop the subscripts; we use $\hat{G}$ to denote the cumulative estimate $\hat{G}_{t-1}$, $\hat{g}$ to denote the marginal estimate $\hat{g}_t = \hat{G}_t - \hat{G}_{t-1}$, and $g$ to denote the true loss $g_t$.

Note that by definition of Algorithm 1, $\hat{g}$ is a sparse vector with one non-zero and non-positive coordinate $\hat{g}_{i_t} = g_{t,i}/\nabla_{i_t}\tilde{\Phi}(\hat{G})$. Plus, $i_t$ is conditionally independent given $\hat{G}$. For a fixed $i_t$, Let

$$h(r) := D_{\tilde{\Phi}}(\hat{G} + r\hat{g}/\|\hat{g}\|, \hat{G}) = D_{\tilde{\Phi}}(\hat{G} + r\mathbf{e}_{i_t}, \hat{G}),$$

so that $h''(r) = (\hat{g}/\|\hat{g}\|)^\top \nabla^2 \tilde{\Phi} \left( \hat{G} + t\hat{g}/\|\hat{g}\| \right) (\hat{g}/\|\hat{g}\|) = \mathbf{e}_{i_t}^\top \nabla^2 \tilde{\Phi} \left( \hat{G} - t\mathbf{e}_{i_t} \right) \mathbf{e}_{i_t}$. Now we can write

$$
\begin{aligned}
\mathbb{E}_{i_t}[D_{\tilde{\Phi}}(\hat{G} + \hat{g}, \hat{G})|\hat{G}] &= \sum_{i=1}^{N} \mathbb{P}[i_t = i] \int_0^{\|\hat{g}\|} \int_0^s h''(r) \, dr \, ds \\
&= \sum_{i=1}^{N} \nabla_i \tilde{\Phi}(\hat{G}) \int_0^{\|\hat{g}\|} \int_0^s \mathbf{e}_i^\top \nabla^2 \tilde{\Phi} \left( \hat{G} - r\mathbf{e}_i \right) \mathbf{e}_i \, dr \, ds \\
&\leq \sum_{i=1}^{N} \nabla_i \tilde{\Phi}(\hat{G}) \int_0^{\|\hat{g}\|} \int_0^s C \left( \nabla_i \tilde{\Phi}(\hat{G} - r\mathbf{e}_i) \right)^\gamma \, dr \, ds \\
&\leq \sum_{i=1}^{N} \nabla_i \tilde{\Phi}(\hat{G}) \int_0^{\|\hat{g}\|} \int_0^s C \left( \nabla_i \tilde{\Phi}(\hat{G}) \right)^\gamma \, dr \, ds \\
&= C \sum_{i=1}^{N} \left( \nabla_i \tilde{\Phi}(\hat{G}) \right)^{1+\gamma} \int_0^{\|\hat{g}\|} \int_0^s \, dr \, ds \\
&= \frac{C}{2} \sum_{i=1}^{N} \left( \nabla_i \tilde{\Phi}(\hat{G}) \right)^{\gamma-1} g_i^2 \leq C \sum_{i=1}^{N} \left( \nabla_i \tilde{\Phi}(\hat{G}) \right)^{\gamma-1}.
\end{aligned}
$$

The first inequality is by the supposition and the second inequality is due to the convexity of $\tilde{\Phi}$ which guarantees that $\nabla_i$ is an increasing function in the $i$-th coordinate. Interestingly, this part of the proof critically depends on the fact that the we are in the "loss" setting where $g$ is always non-positive. $\qquad\square$

## 3 A Minimax Bandit Algorithm via Tsallis Smoothing

The design of a multi-armed bandit algorithm in the adversarial setting proved to be a challenging task. Ignoring the dependence on $N$ for the moment, we note that the initial published work on EXP3 provided only an $O(T^{2/3})$ guarantee (Auer et al., 1995), and it was not until the final version of this work (Auer et al., 2003) that the authors obtained the optimal $O(\sqrt{T})$ rate. For the more

general setting of online linear optimization, several sub-optimal rates were achieved (Dani and Hayes, 2006; Flaxman et al., 2005; McMahan and Blum, 2004) before the desired $\sqrt{T}$ was obtained (Abernethy et al., 2012; Dani et al., 2008).

We can view EXP3 as an instance of GBPA where the potential function $\tilde{\Phi}(\cdot)$ is the Fenchel conjugate of the *Shannon entropy*. For any $p \in \Delta_N$, the (negative) Shannon entropy is defined as $H(p) := \sum_i p_i \log p_i$, and its Fenchel conjugate is $H^\star(G) = \sup_{p \in \Delta_N} \{\langle p, G \rangle - \eta H(p)\}$. In fact, we have a closed-form expression for the supremum: $H^\star(G) = \frac{1}{\eta} \log\left(\sum_i \exp(\eta G_i)\right)$. By inspecting the gradient of the above expression, it is easy to see that EXP3 chooses the distribution $p_t = \nabla H^\star(G)$ every round.

The tighter EXP3 bound given by Auer et al. (2003) scaled according to $O(\sqrt{TN \log N})$ and the authors provided a matching lower bound of the form $\Omega(\sqrt{TN})$. It remained an open question for some time whether there exists a minimax optimal algorithm that does not contain the log term until Audibert and Bubeck (2009) proposed the Implicitly Normalized Forecaster (INF). The INF is implicitly defined via a specially-designed potential function with certain properties. It was not immediately clear from this result how to define a minimax-optimal algorithm using the now-standard tools of regularization and Bregman divergence.

More recently, Audibert et al. (2011) improved upon Audibert and Bubeck (2009), extending the results to the combinatorial setting, and they also discovered that INF can be interpreted in terms of Bregman divergences. We give here a reformulation of INF that leads to a very simple analysis in terms of our notion of *differential consistency*. Our reformulation can be viewed as a variation of EXP3, where the key modification is to replace the Shannon entropy function with the *Tsallis entropy*[1] for parameter $0 < \alpha < 1$:

$$S_\alpha(p) = \frac{1}{1-\alpha}\left(1 - \sum p_i^\alpha\right).$$

This particular function, proposed by Tsallis (1988), possesses a number of natural properties. The Tsallis entropy is in fact a generalization of the Shannon entropy, as one obtains the latter as a special case of the former asymptotically. That is, it is easy to prove the following uniform convergence:

$$S_\alpha(\cdot) \to H(\cdot) \qquad \text{as } \alpha \to 1.$$

We emphasize again that one can easily show that Tsallis-smoothing bandit algorithm is indeed *identical* to INF using the appropriate parameter mapping, although our analysis is simpler due to the notion of differential consistency (Definition 2.2).

**Theorem 3.1.** *Let* $\tilde{\Phi}(G) = \max_{p \in \Delta_N} \{\langle p, G \rangle - \eta S_\alpha(p)\}$. *Then the GBPA($\tilde{\Phi}$) has regret at most*

$$\mathbb{E}\text{Regret} \leq \eta \frac{N^{1-\alpha} - 1}{1 - \alpha} + \frac{N^\alpha T}{\eta \alpha}. \tag{5}$$

Before proving the theorem, we note that it immediately recovers the EXP3 upper bound as a special case $\alpha \to 1$. An easy application of L'Hôpital's rule shows that as $\alpha \to 1$, $\frac{N^{1-\alpha}-1}{1-\alpha} \to \log N$ and $N^\alpha/\alpha \to N$. Choosing $\eta = \sqrt{(N \log N)/T}$, we see that the right-hand side of (5) tends to $2\sqrt{TN \log N}$. However the choice $\alpha \to 1$ is clearly not the optimal choice, as we show in the following statement, which directly follows from the theorem once we see that $N^{1-\alpha} - 1 < N^{1-\alpha}$.

**Corollary 3.2.** *For any* $\alpha \in (0, 1)$, *if we choose* $\eta = \sqrt{\frac{\alpha N^{1-2\alpha}}{(1-\alpha)T}}$ *then we have*

$$\mathbb{E}\text{Regret} \leq 2\sqrt{\frac{NT}{\alpha(1-\alpha)}}.$$

*In particular, the choice of* $\alpha = \frac{1}{2}$ *gives a regret of no more than* $4\sqrt{NT}$.

*Proof of Theorem 3.1.* We will bound each penalty term in Lemma 2.1. Since $S_\alpha$ is non-positive, the **underestimation penalty** is upper bounded by 0 and the **overestimation penalty** is at most $(-\min S_\alpha)$. The minimum of $S_\alpha$ occurs at $(1/N, \ldots, 1/N)$. Hence,

$$(\text{overestimation penalty}) \leq -\frac{\eta}{1-\alpha}\left(1 - \sum_{i=1}^N \frac{1}{N^\alpha}\right) \leq \eta(N^{1-\alpha} - 1). \tag{6}$$

Now it remains to upper bound the **divergence penalty** with $(\eta\alpha)^{-1}N^{\alpha}T$. We observe that straightforward calculus gives $\nabla^2 S_\alpha(p) = \eta\alpha\,\mathrm{diag}(p_1^{\alpha-2}, \ldots, p_N^{\alpha-2})$. Let $\mathbb{I}_{\Delta_N}(\cdot)$ be the indicator function of $\Delta_N$; that is, $\mathbb{I}_{\Delta_N}(x) = 0$ for $x \in \Delta_N$ and $\mathbb{I}_{\Delta_N}(x) = \infty$ for $x \notin \Delta_N$. It is clear that $\tilde{\Phi}(\cdot)$ is the dual of the function $S_\alpha(\cdot) + \mathbb{I}_{\Delta_N}(\cdot)$, and moreover we observe that $\nabla^2 S_\alpha(p)$ is a *sub-hessian* of $S_\alpha(\cdot) + \mathbb{I}_{\Delta_N}(\cdot)$ at $p(G)$, following the setup of Penot (1994). Taking advantage of Proposition 3.2 in the latter reference, we conclude that $\nabla^{-2} S_\alpha(p(G))$ is a *super-hessian* of $\tilde{\Phi} = S_\alpha^*$ at $G$. Hence,

$$\nabla^2 \tilde{\Phi}(G) \preceq (\eta\alpha)^{-1}\mathrm{diag}(p_1^{2-\alpha}(G), \ldots, p_N^{2-\alpha}(G))$$

for any $G$. What we have stated, indeed, is that $\tilde{\Phi}$ is $(2 - \alpha, (\eta\alpha)^{-1})$-differentially-consistent, and thus applying Theorem 2.3 gives

$$D_{\tilde{\Phi}}(\hat{G}_t, \hat{G}_{t-1}) \leq (\eta\alpha)^{-1} \sum_{i=1}^{N} \left( p_i(\hat{G}_{t-1}) \right)^{1-\alpha}.$$

Noting that the $\frac{1}{\alpha}$-norm and the $\frac{1}{1-\alpha}$-norm are dual to each other, we can apply Hölder's inequality to any probability distribution $p_1, \ldots, p_N$ to obtain

$$\sum_{i=1}^{N} p_i^{1-\alpha} = \sum_{i=1}^{N} p_i^{1-\alpha} \cdot 1 \leq \left( \sum_{i=1}^{N} p_i^{\frac{1-\alpha}{1-\alpha}} \right)^{1-\alpha} \left( \sum_{i=1}^{N} 1^{\frac{1}{\alpha}} \right)^{\alpha} = (1)^{1-\alpha} N^{\alpha} = N^{\alpha}.$$

So, the divergence penalty is at most $(\eta\alpha)^{-1}N^{\alpha}$, which completes the proof. $\qquad\square$

## 4 Near-Optimal Bandit Algorithms via Stochastic Smoothing

Let $\mathcal{D}$ be a continuous distribution over an unbounded support with probability density function $f$ and cumulative density function $F$. Consider the GBPA($\tilde{\Phi}(G; \mathcal{D})$) where

$$\tilde{\Phi}(G; \mathcal{D}) = \mathbb{E}_{Z_1, \ldots, Z_N \overset{\text{iid}}{\sim} \mathcal{D}} \max_i \{G_i + Z_i\}$$

which is a *stochastic smoothing* of $(\max_i G_i)$ function. Since the max function is convex, $\tilde{\Phi}$ is also convex. By Bertsekas (1973), we can swap the order of differentiation and expectation:

$$\tilde{\Phi}(G; \mathcal{D}) = \mathbb{E}_{Z_1, \ldots, Z_N \overset{\text{iid}}{\sim} \mathcal{D}} e_{i^*}, \text{ where } i^* = \underset{i=1,\ldots,N}{\arg\max}\{G_i + Z_i\}. \tag{7}$$

Even if the function is not differentiable everywhere, the swapping is still possible with any subgradient as long as they are bounded. Hence, the ties between coordinates (which happen with probability zero anyways) can be resolved in an arbitrary manner. It is clear that $\nabla\tilde{\Phi}$ is in the probability simplex, and note that

$$\frac{\partial \tilde{\Phi}}{\partial G_i} = \mathbb{E}_{Z_1, \ldots, Z_N} \mathbf{1}\{G_i + Z_i > G_j + Z_j, \forall j \neq i\}$$

$$= \mathbb{E}_{\tilde{G}_{j^*}}[\mathbb{P}_{Z_i}[Z_i > \tilde{G}_{j^*} - G_i]] = \mathbb{E}_{\tilde{G}_{j^*}}[1 - F(\tilde{G}_{j^*} - G_i)] \tag{8}$$

where $\tilde{G}_{j^*} = \max_{j \neq i} G_j + Z_j$. The unbounded support condition guarantees that this partial derivative is non-zero for all $i$ given any $G$. So, $\tilde{\Phi}(G; \mathcal{D})$ satisfies the requirements of Algorithm 1.

### 4.1 Connection to Follow the Perturbed Leader

There is a straightforward way to efficiently implement the sampling step of the bandit GBPA (Algorithm 1) with a stochastically smoothed function. Instead of evaluating the expectation of Equation 7, we simply take a random sample. In fact, this is equivalent to **Follow the Perturbed Leader Algorithm (FTPL)** (Kalai and Vempala, 2005) for bandit settings. On the other hand, implementing the estimation step is hard because generally there is no closed-form expression for $\nabla\tilde{\Phi}$.

To address this issue, Neu and Bartók (2013) proposed Geometric Resampling (GR). GR uses an iterative resampling process to estimate $\nabla_i\tilde{\Phi}$. This process gives an unbiased estimate when allowed

to run for an unbounded number of iterations. Even when we truncate the resampling process after $M$ iterations, the extra regret due to the estimation bias is at most $\frac{NT}{eM}$ (additive term). Since the lower bound for the multi-armed bandit problem is $O(\sqrt{NT})$, any choice of $M = O(\sqrt{NT})$ does not affect the asymptotic regret of the algorithm. In summary, all our GBPA regret bounds in this section hold for the corresponding FTPL algorithm with an extra additive $\frac{NT}{eM}$ term in the bound.

Despite the fact that perturbation-based algorithms provide a natural randomized decision strategy, they have seen little applications mostly because they are hard to analyze. But one should expect general results to be within reach: the EXP3 algorithm, for example, can be viewed through the lens of perturbations, where the noise is distributed according to the Gumbel distribution. Indeed, an early result of Kujala and Elomaa (2005) showed that a near-optimal MAB strategy comes about through the use of exponentially-distributed noise, and the same perturbation strategy has more recently been utilized in the work of Neu and Bartók (2013) and Kocák et al. (2014). However, a more general understanding of perturbation methods has remained elusive. For example, would Gaussian noise be sufficient for a guarantee? What about, say, the Weibull distribution?

## 4.2 Hazard Rate analysis

In this section, we show that the performance of the GBPA($\tilde{\Phi}(G; \mathcal{D})$) can be characterized by the *hazard function* of the smoothing distribution $\mathcal{D}$. The hazard rate is a standard tool in survival analysis to describe failures due to aging; for example, an increasing hazard rate models units that deteriorate with age while a decreasing hazard rate models units that improve with age (a counter intuitive but not illogical possibility). To the best of our knowledge, the connection between hazard rates and design of adversarial bandit algorithms has not been made before.

**Definition 4.1** (Hazard rate function)**.** *Hazard rate function of a distribution $\mathcal{D}$ is*

$$h_\mathcal{D}(x) := \frac{f(x)}{1 - F(x)}$$

For the rest of the section, we assume that $\mathcal{D}$ is unbounded in the direction of $+\infty$, so that the hazard function is well-defined everywhere. This assumption is for the clarity of presentation and can be easily removed (Appendix B).

**Theorem 4.2.** *The regret of the GBPA on $\tilde{\Phi}(L) = \mathbb{E}_{Z_1,\ldots,Z_n \sim D} \max_i\{G_i + \eta Z_i\}$ is at most:*

$$\frac{N(\sup h_\mathcal{D})}{\eta}T + \eta \mathbb{E}_{Z_1,\ldots,Z_n \sim D}\left[\max_i Z_i\right]$$

*Proof.* We analyze each penalty term in Lemma 2.1. Due to the convexity of $\Phi$, the underestimation penalty is non-positive. The overestimation penalty is clearly at most $\mathbb{E}_{Z_1,\ldots,Z_n \sim D}[\max_i Z_i]$, and Lemma 4.3 proves the $N(\sup h_\mathcal{D})$ upper bound on the divergence penalty.

It remains to provide the tuning parameter $\eta$. Suppose we scale the perturbation $Z$ by $\eta > 0$, i.e., we add $\eta Z_i$ to each coordinate. It is easy to see that $\mathbb{E}[\max_{i=1,\ldots,n} \eta X_i] = \eta \mathbb{E}[\max_{i=1,\ldots,n} X_i]$. For the divergence penalty, let $F_\eta$ be the CDF of the scaled random variable. Observe that $F_\eta(t) = F(t/\eta)$ and thus $f_\eta(t) = \frac{1}{\eta}f(t/\eta)$. Hence, the hazard rate scales by $1/\eta$, which completes the proof. □

**Lemma 4.3.** *The divergence penalty of the GBPA with $\tilde{\Phi}(G) = \mathbb{E}_{Z \sim \mathcal{D}} \max_i\{G_i + Z_i\}$ is at most $N(\sup h_\mathcal{D})$ each round.*

*Proof.* Recall the gradient expression in Equation 8. The $i$-th diagonal entry of the Hessian is:

$$\nabla^2_{ii}\tilde{\Phi}(G) = \frac{\partial}{\partial G_i}\mathbb{E}_{\tilde{G}_{j^*}}[1 - F(\tilde{G}_{j^*} - G_i)] = \mathbb{E}_{\tilde{G}_{j^*}}\left[\frac{\partial}{\partial G_i}(1 - F(\tilde{G}_{j^*} - G_i))\right] = \mathbb{E}_{\tilde{G}_{j^*}}f(\tilde{G}_{j^*} - G_i)$$

$$= \mathbb{E}_{\tilde{G}_{j^*}}[h(\tilde{G}_{j^*} - G_i)(1 - F(\tilde{G}_{j^*} - G_i))] \tag{9}$$

$$\leq (\sup h)\mathbb{E}_{\tilde{G}_{j^*}}[1 - F(\tilde{G}_{j^*} - G_i)]$$

$$= (\sup h)\nabla_i(G)$$

where $\tilde{G}_{j^*} = \max_{j \neq i}\{G_j + Z_j\}$ which is a random variable independent of $Z_i$. We now apply Theorem 2.3 with $\gamma = 1$ and $C = (\sup h)$ to complete the proof. □

| Distribution | $\sup_x h_{\mathcal{D}}(x)$ | $\mathbb{E}[\max_{i=1}^N Z_i]$ | $O(\sqrt{TN\log N})$ Param. |
|---|---|---|---|
| Gumbel($\mu=1, \beta=1$) | 1 as $x \to 0$ | $\log N + \gamma_0$ | N/A |
| Frechet ($\alpha > 1$) | at most $2\alpha$ | $N^{1/\alpha}\Gamma(1-1/\alpha)$ | $\alpha = \log N$ |
| Weibull*($\lambda=1, k \leq 1$) | $k$ at $x=0$ | $O((\frac{1}{k})!(\log N)^{\frac{1}{k}})$ | $k=1$ (Exponential) |
| Pareto*($x_m=1, \alpha$) | $\alpha$ at $x=0$ | $\alpha N^{1/\alpha}/(\alpha-1)$ | $\alpha = \log N$ |
| Gamma($\alpha \geq 1, \beta$) | $\beta$ as $x \to \infty$ | $\log N + (\alpha-1)\log\log N - \log\Gamma(\alpha) + \beta^{-1}\gamma_0$ | $\beta = \alpha = 1$ (Exponential) |

Table 1: *Distributions that give $O(\sqrt{TN\log N})$ regret FTPL algorithm.* The parameterization follows Wikipedia pages for easy lookup. We denote the Euler constant ($\approx 0.58$) by $\gamma_0$. Distributions marked with (*) need to be slightly modified using the conditioning trick explained in Appendix B.2. The maximum of Frechet hazard function has to be computed numerically (Elsayed, 2012, p. 47) but elementary calculations show that it is bounded by $2\alpha$ (Appendix D).

**Corollary 4.4.** *Follow the Perturbed Leader Algorithm with distributions in Table 1 (restricted to a certain range of parameters), combined with Geometric Resampling (Section 4.1) with $M = \sqrt{NT}$, has an expected regret of order $O(\sqrt{TN\log N})$.*

Table 1 provides the two terms we need to bound. We derive the third column of the table in Appendix C using Extreme Value Theory (Embrechts et al., 1997). Note that our analysis in the proof of Lemma 4.3 is quite tight; the only place we have an inequality is when we upper bound the hazard rate. It is thus reasonable to pose the following conjecture:

**Conjecture 4.5.** *If a distribution $\mathcal{D}$ has a monotonically increasing hazard rate $h_{\mathcal{D}}(x)$ that does not converge as $x \to +\infty$ (e.g., Gaussian), then there is a sequence of losses that will incur at least a linear regret.*

The intuition is that if adversary keeps incurring a high loss for the $i$-th arm, then with high probability $\tilde{G}_{j^*} - G_i$ will be large. So, the expectation in Equation 9 will be dominated by the hazard function evaluated at large values of $\tilde{G}_{j^*} - G_i$.

**Acknowledgments.** J. Abernethy acknowledges the support of NSF under CAREER grant IIS-1453304. A. Tewari acknowledges the support of NSF under CAREER grant IIS-1452099.

## Footnotes

[1]More precisely, the function we give here is the *negative* Tsallis entropy according to its original definition.

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
