[Supplementary Material]

---

**Algorithm 2:** Gradient-Based Prediction Algorithm (GBPA) for Full Information Setting

---

Input: $\tilde{\Phi}$, a differentiable convex function such that $\nabla \tilde{\Phi} \in \Delta^N$ and $\nabla_i \tilde{\Phi} > 0$ for all $i$.
Initialize $G_0 = 0$
**for** $t = 1$ *to T* **do**

    **Sampling:** The learner chooses arm $i_t$ with probability $p_i(\hat{G}_{t-1}) = \nabla_i \Phi_t(\hat{G}_{t-1})$
    Adversary chooses a loss vector $g_t \in [-1,0]^N$ and learner pays $g_{t,i}$
    Update $\hat{G}_t = G_{t-1} + g_t$

---

## A   Proof of the GBPA Regret Bound (Lemma 2.1)

**Lemma A.1.** *The expected regret of Algorithm 2 can be written as:*

$$\mathbb{E}\text{Regret} = \underbrace{\tilde{\Phi}(0) - \Phi(0)}_{\textit{overestimation penalty}} + \underbrace{\Phi(G_T) - \tilde{\Phi}(G_T)}_{\textit{underestimation penalty}} + \sum_{t=1}^{T} \underbrace{D_{\tilde{\Phi}}(G_t, G_{t-1})}_{\textit{divergence penalty}}$$

*Proof.* Note that since $\Phi_0(0) = 0$,

$$\tilde{\Phi}(G_T) = \underbrace{\left(\tilde{\Phi}(0) - \Phi_0(0)\right)}_{\text{overestimation penalty}} + \sum_{t=1}^{T} \tilde{\Phi}(G_t) - \tilde{\Phi}(G_{t-1})$$

$$= \underbrace{\left(\tilde{\Phi}(0) - \Phi_0(0)\right)}_{\text{overestimation penalty}} + \sum_{t=1}^{T} \langle \nabla \tilde{\Phi}(G_{t-1}), \ell_t \rangle + D_{\tilde{\Phi}}(G_t, G_{t-1})$$

Therefore,

$$\mathbb{E}\text{Regret} \overset{\text{def}}{=} \mathbb{E}\left[ \Phi(G_T) - \sum_{t=1}^{T} \langle \tilde{\Phi}(G_{t-1}), g_t \rangle \right]$$

$$= \mathbb{E}\left[ \underbrace{\left(\Phi(G_T) - \tilde{\Phi}(G_T)\right)}_{\text{underestimation penalty}} + \tilde{\Phi}(G_T) - \sum_{t=1}^{T} \langle \tilde{\Phi}(G_{t-1}), g_t \rangle \right]$$

$$= \mathbb{E}\left[ \underbrace{\left(\Phi(G_T) - \tilde{\Phi}(G_T)\right)}_{\text{underestimation penalty}} + \underbrace{\left(\tilde{\Phi}(0) - \Phi_0(0)\right)}_{\text{overestimation penalty}} + D_{\tilde{\Phi}}(G_t, G_{t-1}) \right]$$

$\square$

## B   Relaxing Assumptions on the Distribution

### B.1   Mirroring trick for extending the support

Let $X$ have support on $x > 0$ with density $f$ and CDF $F$. Let us define $Y$ by mirroring the density of $X$ around zero, i.e., $Y$ has density $g(y) = \frac{1}{2}f(|y|)$ and CDF $G(y) = \frac{1}{2}\left(1 + \text{sign}(y)F(|y|)\right)$. Note that $|Y|$ is distributed as $X$ and hence,

$$\mathbb{E}[\max_i Y_i] \le \mathbb{E}[\max_i |Y_i|] = \mathbb{E}[\max_i X_i].$$

The hazard $h_Y(y)$ for $y \ge 0$ is $f(y)/(1-F(y))$ and for $y < 0$ is $f(-y)/(1+F(-y)) \le F(-y)/(1-F(-y))$. Therefore,

$$\sup_y h_Y(y) = \sup_{x>0} h_X(x).$$

This proves the following lemma.

**Lemma B.1.** *If a random variable $X$ has support on the non-negative reals with density $f(x)$ and we define $Y$ as the mirrored version with density $g(y) = \frac{1}{2} f(|y|)$. Then, we have*

$$\mathbb{E}[\max_i Y_i] \leq \mathbb{E}[\max_i X_i],$$

$$\sup_y h_Y(y) = \sup_{x>0} h_X(x)$$

*where $h_X, h_Y$ are hazard rates of $X, Y$ respectively.*

## B.2 Conditioning trick for unbounded hazard rate near zero

Suppose $F(x)$ is the CDF of a random variable $X$ whose hazard rate is bounded for $x \geq 1$ but blows up near zero. Then define $Y$ as $X$ conditioned on $X \geq 1$. That is, $Y$ has CDF, for $y > 0$:

$$G(y) = P(X \geq 1 + y | X > 1) = \frac{F(1+y) - F(1)}{1 - F(1)}$$

and density $g(y) = f(1+y)/(1 - F(1)), y > 0$. So the hazard rate $h_Y(y)$ is $g(y)/(1 - G(y)) = f(1+y)/(1 - F(1+y)) = h_X(1+y)$. Therefore,

$$\sup_{y>0} h_Y(y) = \sup_{x>1} h_X(x)$$

which makes the hazard rate of $Y$ now bounded. This we have proved the lemma below.

**Lemma B.2.** *If a hazard function of $X$ is bounded for $x > 1$ and blows up only for small values of $x$ then we can condition on $X > 1$ to define a new random variable whose hazard rate is now bounded.*

The same technique can be applied for any arbitrary constant other than 1, but for the family of random variables we considered, it suffices to condition on $X \geq 1$.

# C Detailed derivation of extreme value behavior

## C.1 Maximum of iid Gumbel

The CDF of the Gumbel distribution is $\exp(-\exp(-x))$ and the expected value is $\gamma_0$, the Euler (Euler-Mascheroni) constant. Thus, the CDF of the maximum of $n$ iid Gumbel random variables is $(\exp(-\exp(-x)))^N = \exp(-\exp(-(x - \log N)))$ which is also Gumbel but with the mean increased by $\log N$.

## C.2 Maximum of iid Frechet

The CDF of Frechet is $\exp(-x^{-\alpha})$ and it has mean $\Gamma(1 - \frac{1}{\alpha})$ as long as $\alpha > 1$ (otherwise it is infinite). Hence, the CDF of the maximum of $N$ iid Frechet random variables is

$$(\exp(-x^{-\alpha}))^N = \exp(-Nx^{-\alpha}) = \exp\left(-\left(\frac{x}{N^{\frac{1}{\alpha}}}\right)^{-\alpha}\right)$$

which is also Frechet but with mean scaled by $N^{1/\alpha}$.

## C.3 Maximum of iid Weibull

Let $X_i$ have modified Weibull distribution with CDF $1 - \exp(-(x+1)^k + 1)$. Thus, $P(\max_i X_i > t) \leq NP(X_1 > t) = N \exp(-(t+1)^k + 1)$. For non-negative random variable $X$ and any $u > 0$, we have,

$$\mathbb{E}[X] = \int_0^\infty P(X > x)dx \leq u + \int_u^\infty P(X > x)dx.$$

Assume $k = 1/m$ where $m \geq 1$ is a positive integer. Therefore,

$$\mathbb{E}[\max_i X_i] \leq u + \int_u^\infty N \exp(-(x+1)^k + 1)dx$$

$$\leq u + 3N \int_u^\infty \exp(-(x+1)^k)dx$$

$$= u + 3N \int_{u+1}^\infty \exp(-x^{1/m})dx$$

$$= u + 3Nm\Gamma(m, (1+u)^{1/m})dx$$

where $\Gamma(m, x)$ is the incomplete Gamma function that for a positive integer $m$ and $x > 1$ simplifies to

$$\Gamma(m, x) = (m-1)!e^{-x} \sum_{k=0}^{m-1} \frac{x^k}{k!} \leq (m-1)!e^{-x} \sum_{k=0}^{m-1} \frac{x^m}{k!}$$

$$= (m-1)!e^{-x}x^m \sum_{k=0}^{m-1} \frac{1}{k!} \leq (m-1)!e^{-x}x^m \sum_{k=0}^{\infty} \frac{1}{k!}$$

$$\leq 3(m-1)!e^{-x}x^m.$$

Plugging this back above, we get, for any $u > 0$,

$$\mathbb{E}[\max_i X_i] \leq u + 9Nm!e^{-(1+u)^{1/m}}(1+u).$$

Now choose $u = \log^m N + 1$ to get

$$\mathbb{E}[\max_i X_i] \leq \log^m N + 9Nm!\frac{\log^m N}{N} \leq 10\,m!\log^m N.$$

### C.4   Maximum of iid Gamma

Let $Y$ be the maximum of $N$ iid Gamma$(\alpha, \beta)$ ramdom variables. Then, $\frac{Y-d_N}{c_N}$ follows Gumbel distribution, where $c_N = \beta^{-1}$ and $d_N = \beta^{-1}(\log N + (\alpha - 1)\log\log N - \log\Gamma(\alpha))$. In the language of extreme value theory, Gamma distribution belongs to the maximum domain of attraction of Gumbel distribution with parameters (Embrechts et al., 1997). As mentioned in Section C.1, Gumbel distribution has mean $\gamma_0$.

### C.5   Maximum of iid Pareto

Let $X_i$ have modified Pareto distribution with CDF $1 - 1/(1+x)^\alpha$. Thus, $P(\max_i X_i > t) \leq NP(X_1 > t) = N/(1+x)^\alpha$. For non-negative random variable $X$ and any $u > 0$, we have,

$$\mathbb{E}[X] = \int_0^\infty P(X > x)dx \leq u + \int_u^\infty P(X > x)dx.$$

Therefore, for $\alpha > 1$,

$$\mathbb{E}[\max_i X_i] \leq u + \int_u^\infty \frac{N}{(1+x)^\alpha}dx$$

$$= u + \frac{N}{(\alpha-1)(1+u)^{\alpha-1}}.$$

Setting $u = N^{1/\alpha} - 1$ gives the bound

$$\mathbb{E}[\max_i X_i] \leq \frac{\alpha}{\alpha-1}N^{1/\alpha}.$$

# D  Hazard Functions of Modified Distributions and the Frechet Case

## D.1  Pareto distribution

Using the conditioning trick, we consider, for $\alpha > 1$ (otherwise mean is infinite), the modified Pareto distribution with pdf $f(x) = \frac{\alpha}{(x+1)^{\alpha+1}}$ supported on $(0, \infty)$. Its CDF is $1 - 1/(x+1)^{\alpha}$. Its hazard function is $h(x) = \frac{\alpha}{x+1}$ which decreases in $x$ and is bounded by $\alpha$. Expected maximum of $N$ iid Pareto random variables is bounded by $\alpha N^{1/\alpha}/(\alpha - 1)$ (see Appendix C.5). This gives a regret bound of $\sqrt{NT}\sqrt{\alpha^2 N^{1/\alpha}/(\alpha - 1)}$.

## D.2  Frechet distribution

The CDF of Frechet is $\exp(-x^{-\alpha}), x > 0$ where $\alpha > 0$ is a shape parameter. The hazard function of Frechet distribution is $h(x) = \alpha x^{-\alpha-1} \frac{\exp(-x^{-\alpha})}{1 - \exp(-x^{-\alpha})}$ which is hard to optimize analytically but can be upper bounded, for $\alpha > 1$, via elementary calculations given below, by $2\alpha$. The CDF of the maximum of $N$ iid Frechet random variables is $\exp(-(x/N^{1/\alpha})^{-\alpha})$ which is also Frechet (but with mean scaled by $N^{1/\alpha}$) with expected value $N^{1/\alpha}\Gamma(1 - \frac{1}{\alpha})$ (as long as $\alpha > 1$, otherwise expectation is infinite). Thus, the regret bound we get is $O\left(\sqrt{NT}\sqrt{\alpha N^{1/\alpha}\Gamma(1 - \frac{1}{\alpha})}\right)$. Setting $\alpha = \log N$ makes the regret bound $O(\sqrt{TN\log N})$. Our choice of $\alpha$ is larger than 1 as soon as $N > 2$.

### D.2.1  Elementary calculations for bounding Frechet distribution's hazard rate

For $\alpha > 1$, we want to show that $\sup_{x>0} h(x) \le 2\alpha$ where

$$h(x) = \alpha x^{-\alpha-1} \frac{\exp(-x^{-\alpha})}{1 - \exp(-x^{-\alpha})}.$$

First, consider the case $x \ge 1$. In this case, define $y = x^{\alpha}$ and note that $y \ge 1$. Then, we have

$$h(x) = \frac{\alpha}{xy} \frac{\exp(-1/y)}{1 - \exp(-1/y)} \le \frac{\alpha}{y} \frac{\exp(-1/y)}{1 - \exp(-1/y)}$$
$$\le \frac{\alpha}{y} \frac{1}{1 - (1 - 1/(2y))} = 2\alpha.$$

The first inequality holds because $x \ge 1$. The second holds because $\exp(-1/y) < 1$ and $\exp(-1/y) \le 1 - 1/(2y)$ for $y \ge 1$.

Next, consider the case $x < 1$. Define $y = 1/x$ and note that $y > 1$. Then, we have

$$h(x) = \frac{\alpha}{x^{\alpha+1}} \frac{\exp(-x^{-\alpha})}{1 - \exp(-x^{-\alpha})} \le \frac{\alpha}{x^{\alpha+1}} \frac{\exp(-x^{-\alpha})}{1 - \exp(-1)}$$
$$= \frac{\alpha}{1 - e^{-1}} y^{\alpha+1} \exp(-y^{\alpha}) \le 2\alpha y^{\alpha+1} \exp(-y^{\alpha}).$$

To show an upper bound of $2\alpha$, it therefore suffices to show that $\sup_{y>1} g(y) \le 1$ where $g(y) = y^{\alpha+1} \exp(-y^{\alpha})$. We will show this now. Note that

$$g'(y) = (\alpha + 1)y^{\alpha} \exp(-y^{\alpha}) - y^{\alpha+1}\alpha y^{\alpha-1} \exp(-y^{\alpha}) = y^{\alpha} \exp(-y^{\alpha})\left((\alpha + 1) - \alpha y^{\alpha}\right),$$

which means that $g(y)$ is monotonically increasing on the interval $(1, y_0)$ and monotonically decreasing on the interval $(y_0, +\infty)$ where $y_0 = \left(\frac{\alpha+1}{\alpha}\right)^{1/\alpha}$. We therefore have,

$$\sup_{y>1} g(y) = g(y_0) = \left(1 + \frac{1}{\alpha}\right)^{(1+1/\alpha)} \exp\left(-(1 + 1/\alpha)\right) \le 2^2 \exp(-2) = 4/e^2 \le 1,$$

where the first inequality above holds because $\alpha > 1$. Note that, for $\alpha > 1$, the function $\alpha \mapsto \left(1 + \frac{1}{\alpha}\right)^{(1+1/\alpha)} \exp\left(-(1 + 1/\alpha)\right)$ decreases monotonically.

### D.3 Weibull distribution

The CDF of Weibull is $1 - \exp(-x^k)$ for $x > 0$ (and $0$ otherwise) where $k > 0$ is a shape parameter. The density is $kx^{k-1}\exp(-x^k)$ and hazard rate is $kx^{k-1}$. For $k > 1$, hazard rate monotonically increases and is therefore unbounded for large $x$. When $k < 1$, the hazard rate is unbounded for small values of $x$. Note that Weibull includes exponential as a special case when $k = 1$.

Let $k = 1/m$ for some positive integer $m \geq 1$ and using the conditioning trick, consider a modified Weibull with CDF $1 - \exp(-(x+1)^k + 1)$. Density is $k(x+1)^{k-1}\exp(-(x+1)^k + 1)$ and hazard is $k(x+1)^{k-1}$ which is bounded by $k$. When $k < 1$ we get tails heavier than the exponential but not as heavy as a Pareto or a Frechet. The expected value of the maximum of $N$ iid (modified) Weibull random variables with parameter $k = 1/m$ scales as $O(m!(\log N)^m)$ (see Appendix C.3). Thus, we get the regret bound $O(\sqrt{NT}\sqrt{m!(\log n)^m})$. Thus, the entire modified Weibull family yields $O(\sqrt{N\mathrm{polylog}(N)}\sqrt{T})$ regret bounds. The best bound is obtained when $m = 1$, i.e. when the Weibull becomes an exponential.