[Reviews · NeurIPS 2015]

Submitted by Assigned_Reviewer_1

The papers studies the adversarial multi-armed bandit problem, in the context of Gradient based methods. Two standard approaches are considered: penalization by a potential function, and stochastic smoothing. For the first method, the authors suggest to use Tsallis-alpha entropy as opposed to Shannon entropy as a potential function. Regret bounds (Theorem 3.1, Corollary 3.2) are derived for the expected regret and show that the value alpha=1/2 is especially interesting. For the second method, the authors suggest a regret bound (Theorem 4.2) depending on the Hazard rate function of the smoothing distribution. A notion of smoothness is introduced in Definition 2.2.

Remarks:

The connection between the two methods and the Definition 2.2 does not appear clearly enough in the paper; Perhaps you can explain this more.

D_Phi is not formally defined in the main text.

I understand from the proof that it is the Bregman divergence associated to the potential Phi.

From the proof of Theorem 3.1/Corollary 3.2, it seems that the key value of alpha = 1/2, which corresponds to gamma = 3/2 in your definition 2.2. This shows a striking analogy with the definition of self-concordant barrier, which can be recovered in case Phi writes Phi = \grad f.

Thus, I would tend to think it may not be necessary to introduce a new concept of differential consistency, and in any case I would prefer to see more connections with self-concordant barriers.

Note also that the Tsallis 1/2 divergence bares strong connection with the Hellinger distance.

Lemma 2.1 : The proof is only stated for L_T, not when using \hat L_T. For the bound to hold, it seems you need to control the error term Phi(L_T) - Phi(\hat L_T) as well.

Page 5, line 225:

For the Fenchel conjugate to match the expression on line 226, you should penalize by H(p) / eta, not by eta*H(p). This bad parametrization also appears in Theorem 3.1 and subsequent expressions. Luckily, this does not affect the results that are generally based on the right equivalent expression (like l. 226). However, this puzzles me and shows the lack of intuition the authors have about these quantities.

L.236: This is actually the negative Tsallis entropy (as mentioned in the footnote).

Theorem 3.1/Corollary 3.2 are not compared against the standard literature. The typical regret is sqrt{2TN\log(N)} (not 2sqrt{TN / log(N)}) thus the provided bound is only interesting when the number of arms satisfies N > e^8. This seems to be a rather poor improvement.

Proof of Theorem 3.1: It seems that you forgot to handle the term Phi(L_T) - Phi(\hat L_T) in the proof.

The hazard rate function already appeared in the study of multi-armed bandits, although the connection with the design of adversarial bandits seems indeed new.

Conjecture: Can you provide some examples of such situations? Just to give people reasons to believe that this conjecture might be true.

Summary: A borderline paper.

Quality: Medium. It seems that Lemma 2.1 needs a fix, and thus subsequent results as well. Clarity: Medium. The papers mixes several ideas without providing much intuition. Originality:

Ok. Significance: Medium. The paper essentially collects standard ideas from the literature, without greatly mentioning which ones are standards nor achieving a big step forward either.

==

The authors' rebuttal answers parts of my concerns. I am ok with the use of Jensen to correct Lemma 2.1. I am still not convinced that the introduction of Definition 2.2 is needed. I however agree to raise my score, given the modifications suggested by the authors.

Note: "Luckily, this does not affect the results", I was referring to the fact that Theorem 3.1 has the right parametrization.

Summary: A confusing paper, that needs to be cleaned-up. The provided results do not seem to improve on the literature and are not discussed enough. It seems some parts of the proof need a fix. I am not convinced that Definition 2.2 is the right notion, and the paper does not provide much intuition about it.

Submitted by Assigned_Reviewer_2

Unfortunately all of Section 3 is already known (and with better constants), see Theorem 4 in Regret in Online Combinatorial Optimization, Math of OR. On the other hand I find Theorem 4.2 quite interesting. It is unfortunate that despite the understanding from Section 3 the authors still cannot obtain an optimal strategy with an appropriate noise distribution. My evaluation of the paper is that it does not meet the bar for NIPS. My opinion would have been different if an optimal noise distribution was proposed, or if Conjecture 4.5 was proved. Another interesting direction to explore would be whether Definition 2.2 can be used in linear bandit setting.
Summary: Half of the paper is already known, the other half is interesting.

Submitted by Assigned_Reviewer_3

The paper considers the classic problem of multi-armed bandits and extends the algorithmic framework of "gradient-based prediction algorithms" (GBPA) to this setting. In this framework, the learner uses a convex potential function to measure its own performance and predicts according to the gradient of the potential function evaluated at the current state. The main result in the current paper is a sufficient condition on the potential function so that it guarantees near-optimal regret bounds in the multi-armed bandit problem, assuming that the learner estimates the losses by the standard importance-weighting technique. Based on this result, two further results are derived: an order-optimal learning algorithm based on running FTRL (Follow-the-Regularized-Leader) with the Tsallis entropy as a regularizer and a sufficient condition for the perturbation distribution in FPL (Follow-the-Perturbed-Leader) to give near-optimal performance guarantees.

The main result and its consequences contribute greatly to the literature on multi-armed bandits: Such an easily verifyable sufficient condition on the regularization functions and the perturbation distributions were highly sought for by many researchers. The contributions of the current paper make the relatively scattered picture of existing results on bandits much clearer. The writing and the technical quality are both excellent, which, together with the strength of the results, make this a very good paper overall.

There are, however, a few minor mistakes that need to be fixed before the paper is published. First, the authors argue that the PolyINF algorithm of Audibert and Bubeck (2009) is "completely different" from gradient-based methods. This, however, is not true: as shown by Audibert, Lugosi and Bubeck (2014, MOR), PolyINF is actually an instance of mirror descent with a polynomial regularizer. Actually, it turns out that the regularizer used by Audibert et al. is the same (up to an additive constant) as the Tsallis entropy as proposed in the current paper. Nevertheless, the analysis presented in the current paper is more elegant in many respects, in that it reveals a close connection between Exp3 and the algorithm giving optimal bounds. The authors also claim an improvement in the best known constant factors, this is actually not true: the leading constant of Audibert et al. is actually better (although this issue can be fixed, see below). All in all, the presentation in the current submission needs to be modified a bit to highlight these connections.

A closer examination of the proofs reveals that the regret bounds presented in the paper can be improved by at least a factor of \sqrt{2}. This factor is lost in the inequality on line 207, where C/2 is upper-bounded by C (and \ell_i^2 is bounded by 1). Propagating this lost factor into Theorems 2.3 and 3.1 improves the bounds derived for Exp3 and PolyINF to match the best known regret bounds for the respective algorithms (see, e.g., the monograph by Bubeck and Cesa-Bianchi, 2012 and the paper of Audibert, Bubeck and Lugosi, 2014).

A minor annoyance in the presentation that while the authors claim to study the loss game, they actually study the gain game with negative gains and aim to maximize the sum of these negative gains. To me, this was often the source of confusion, as a loss should always be intuitively minimized and not maximized as often referred to in the paper (e.g., in Eq. 3 and the following paragraphs). I suggest that the authors refer to gains throughout the paper, reminding the reader that the proofs only work for non-positive gains. (Alternatively, conduct the analysis with non-negative losses. However, this seems to be rather tedious as it gives rise to a bunch of confusing minus signs everywhere in the paper.)

Detailed comments ----------------- 087: "and associated bound" -> "and the associated bound"? 088: "That is, noise must satisfy" -> "Precisely, we show that the noise must satisfy"? 107: "Nature's choices [...] are assumed deterministic throughout." -- Can you clarify this? Does this mean that nature fixes the loss vectors in the beginning of the game (i.e., obliviously) or can be a deterministic function of the past plays of the learner? Is there any particular reason to make this assumption? 119: What does "chosen (but fixed)" precisely mean? 122: What is i here? 126: This would be the right time to note that PolyINF is also closely related. (Actually, it is not FTRL but mirror descent. I think it should be easy to show that for this particular regularizer, MD and FTRL are identical on the simplex, as can be done for the negative Shannon entropy.) 132: I mostly buy this argument about the fluctuations of the loss estimates, but note that the estimates need not be unbiased for good performance. In fact, algorithms that give high-probability bounds all use biased estimates. Also, "O(1/p)" should be "\Theta(1/p)". 156: This proof is a bit too sketchy, and referring to "GBPA run without the estimation step" is rather confusing. Besides Abernethy et al. (2012), Jan Poland's work "FPL analysis for adaptive bandits" (2005, TCS tech. report) also does a great job in explaining this step (and is also a relevant reference for following the perturbed leader in bandit problems). 161: "i_1,...,i_t" -> "i_1,...,i_{t-1}"? 170: "hessian" -> "Hessian" 184: "Then divergence penalty" -> "Then, the divergence penalty"? 192: "i_t is conditionally independent given \hat{L}" -- I am not sure what you mean here. I think it should be more important to note here that i_t is distributed as \nabla\tilde\Phi(\hat{L}). Also, some hats and tilde's are missing in this line. 218: "and it was not until [...] did the authors obtain" -> "and it was not until [...] that the authors obtained"? 225 vs 244: sup vs max? 268: \eta is not part of the regularizer and should not show up here. On the other hand, \tilde{\Phi} should be defined as \eta S + I instead of S + I for everything to check out. 290: This section could use a short introduction to be on par with the previous one. 300: The passage about swapping the order of differentiation and integration comes before differentiation is mentioned at all. Also, \tilde\Phi should be replaced by \nabla\tilde\Phi here. 304 and 305: ">" -> "\ge" would be more consistent with the final expression. Also, technically this expression is a subgradient, as for some specific perturbation distributions, a tie between arms may arise with positive probability. 350: I suppose that "(sup h_D)" refers to the supremum of h_D over its domain, but this is not entirely clear (and Appendix B seems to suggest otherwise). 451 and 477: missing years. 473 and 475: "Boltzmann-Gibbs" and "Monte Carlo". 497 and 500: missing parens. 506 and 509: missing \nabla from the last terms.

bonus remark: I don't like the title -- I first thought that "Fighting bandits" is yet another derivative bandit setting. "A new kind of smoothness" sounds good though.
Summary: A great paper giving new key insights into the multi-armed bandit problem. Very well executed.

Author Feedback
Author rebuttal: We thank all reviewers for carefully reading our paper and providing a lot of helpful comments. We have divided our response into small sections targeting common criticisms and concerns.

CONTRIBUTIONS

At least two of the reviewers observed that our Theorem 3.1 result was "known". It is true that we are matching an existing result, and indeed we learned soon after submitting that Audibert/Bubeck/Lugosi had posed a regularization-like view of the INF algorithm which had the same form as our function. We regret that we totally missed this, and overstated the improvement in the bound, and we intend to correct both errors.

However, the reviewers should note that Theorem 3.1 was not intended as a new groundbreaking result, we were merely trying to emphasize that (near) optimal bandit algorithms can be designed under a certain smoothness condition, and the Tsallis-entropy trick obtains this smoothness. Moreover, viewing INF through the lens of Tsallis smoothing is kind of a neat, since the Tsallis entropy is something of a "generalization" of Shannon entropy. We would also note that our analysis is quite simple and provides a unified proof framework, even if the algorithm had been previously published.

We thank Reviewer 2 and 3 for the reference. We thank Reviewer 2 for his quick proof on how to match the constant factors, this will certainly be included with an anonymous acknowledgment.
As far as Reviewer 1's comment on how our bounds compare to others, a quick answer is that our Theorem 3.1 and Corollary 3.2 will match the previously known best bound (thank to Reviewer 2), which is minimax optimal up to constant factors.

CORRECTNESS OF LEMMA 2.1

On reviewer 1's concern on the validity of Lemma 2.1: Appendix A proves that the RHS of Eq. (4) is equal to $\Phi(\hat L_T) - <\tilde \Phi(\hat L_T), \hat \ell_t>$. So, as reviewer 1 mentions, it's missing $\Phi( L_T) - E[\Phi(\hat L_T)]$, which by Jensen's inequality is always non-positive. So all our *upper bounds* are still valid.
Our argument that we can use (Abernethy et al., 2012, Section V) to prove equality, however, is wrong. The original argument applies to regret with respect to a fixed $u$ that does not depend on the sampled loss sequence $(\hat \ell_t)$, whereas our regret is defined with respect to a potential function $\Phi$ at $\hat L_T$, which adapts to the sampled loss sequence. We thank Reviewer 1 for carefully reading our paper and pointing out this very important error (which, as we said, does NOT affect the regret upper bound).

CORRECTNESS, MISC.

We thank the Reviewer 1 for finding the wrong parameterization on Line 225; it is a bad mistake that we will fix. However, we feel that it is unfair to claim the lack of our intuition based on this easy-to-make mistake, because Thm 3.1 does NOT have a bad parameterization. Large $\eta$ implies a large stability term, which gives a small divergence penalty proportional to $\eta^{-1}$ and a large overestimation penalty proportional to $\eta$.

CLARITY, DEFINITION 2.2

Lem 2.1 gives a generic regret bound for Framework 1, which can be considered as mirror-descent on a potential function $\Phi$. Among the three penalty terms, the divergence penalty term can be bounded using Theorem 2.3, if $\Phi$ satisfies the Definition 2.2.

THE CONJECTURE

Reviewer 1 and 2 pointed out we should expand on our conjecture. We did some preliminary work on proving that Gaussian bandit is suboptimal, but not mature enough to put it in the submission. We will be happy to outline a possible strategy to prove the conjecture in the camera-ready version. However, the absence of a negative result about Gaussian perturbations should not lessen the value of our positive results for a variety of perturbations (Gumbel, Frechet, Weibull, Pareto, Gamma) some of which were never even considered before in the adversarial bandit literature.